# Early Identification of Unbalanced Freight Traffic Loads Based on Wayside Monitoring and Artificial Intelligence

**DOI:** 10.3390/s23031544

**Published:** 2023-01-31

**Authors:** R. Silva, A. Guedes, D. Ribeiro, C. Vale, A. Meixedo, A. Mosleh, P. Montenegro

**Affiliations:** 1CONSTRUCT-LESE, School of Engineering, Polytechnic of Porto, 4249-015 Porto, Portugal; 2CONSTRUCT-LESE, Faculty of Engineering, University of Porto, 4249-015 Porto, Portugal

**Keywords:** unbalanced vertical loads, freight traffic loads, wayside condition monitoring, train–track interaction, artificial intelligence

## Abstract

The identification of instability problems in freight trains circulation such as unbalanced loads is of particular importance for railways management companies and operators. The early detection of unbalanced loads prevents significant damages that may cause service interruptions or derailments with high financial costs. This study aims to develop a methodology capable of automatically identifying unbalanced vertical loads considering the limits proposed by the reference guidelines. The research relies on a 3D numerical simulation of the train–track dynamic response to the presence of longitudinal and transverse scenarios of unbalanced vertical loads and resorting to a virtual wayside monitoring system. This methodology is based on measured data from accelerometers and strain gauges installed on the rail and involves the following steps: (i) feature extraction, (ii) features normalization based on a latent variable method, (iii) data fusion, and (iv) feature discrimination based on an outlier and a cluster analysis. Regarding feature extraction, the performance of ARX and PCA models is compared. The results prove that the methodology is able to accurately detect and classify longitudinal and transverse unbalanced loads with a reduced number of sensors.

## 1. Introduction

Rail transport has increased in the last decade, and it is likely to further increase not only in terms of passenger transportation but also in terms of freight transportation. The shift from road and air to rail is an important demand due to the rising energy costs, congestion of roads and sky, and the demand for reducing gas emissions. Vehicle running safety is a major concern for railway administrations, as a derailment may cause high financial costs [1]. Moreover, the cost-effectiveness of track maintenance and renewal activities should be considered, and for that, track degradation and damage need to be controlled [2]. The presence of unbalanced loads in the track induces damage in the track components, such as rails, sleepers, and ballast, which has implications in maintenance periodicity [3]; thus, detection of these instabilities is of major importance. Condition-based monitoring systems aim to contribute to these two aspects: safety and optimization of maintenance actions. The identification of instability problems in operation is of particular importance for railways management companies and operators by increasing the reliability of this mode of transport and ensuring safety during rail traffic.

Unbalanced loads are an instability problem affecting specially freight trains as they have the potential for causing vehicle derailment. As important as not exceeding the overload limit, the load should be properly distributed in the wagons so that neither the axle nor wheel is overloaded. It is generally accepted that the ideal location of the cargo gravity centre is laterally and longitudinally at the centre of the vehicle. Despite not existing universal requirements on the mass distribution for a loaded wagon, loading guidelines of several rail organizations specified allowable offset values [4,5,6]. This is the case of UIC code [6] which presents limits for the longitudinal and lateral offset of the payload centre of mass.

An uneven mass distribution could result in an obvious unbalance of wheel load and deteriorate the curving performance seriously, as shown by Zhang et al. [7,8] in studies about the effects of mass distribution in curved tracks. According to Pan et al. [9], if trains remain overloaded or unbalanced loaded for a long period of time, the wheels and axles will be subjected to fatigue phenomena that can cause axle breaking, axle cutting, derailment, train subversion, etc. Pagaimo et al. [10] conclude that lateral unbalance of the cargo is one of the most adverse scenarios in studying the potential derailment of freight trains. Moreover, the reports presented by the Rail Accident Investigation Branch in the UK [11] confirm the same conclusion.

Currently, there are several commercial weigh-in-motion systems (WIM), composed of a series of sensors designed to capture the dynamic vertical forces applied by the passage of a vehicle over the rail [3,12,13,14,15]. Mosleh et al. [13] proposed a method to obtain the weigh-in-motion of the train, considering different speeds. Moreover, their study showed how train speed and track unevenness affect the loads assessed by the WIM system. In the study of Pintao et al. [3], a WIM algorithm developed for ballasted tracks is proposed and validated with synthetic data from trains that run in the Portuguese railway network. The proposed methodology to estimate the wheel static load is successfully accomplished, as the load falls within the confidence interval. Although very widespread and efficient, WIM systems are more focused on identifying situations of vehicle weighing and overloading [16,17,18,19], and reported literature on systems capable of detecting situations of instability in railway circulation, such as unbalanced loads, has been limited so far.

Most of the unbalanced systems proposed in the literature are on-board, with some examples found where the detection devices are mounted on vehicle carbody [20], bogies [21], or suspensions [22]. Although the high precision for monitoring unbalance loads of these devices, they require a high installation cost in order to study a wider range of vehicles. The sensors on the railway track allows extracting a lot of data referring to all circulating vehicles and thus enabling the methodology to monitor various types of vehicles with a reduced number of sensors. For example, Alotta et al. [23] show a WIM-based algorithm to verify all types of loading conditions of a wide variety of vehicles using strain gauges. Moreover, Pan et al. [9] present an application of Fibre Bragg Grating (FBG) strain gauges for train overload and unbalanced load monitoring, by comparing the dynamic wheel axle load values on the two rails. Additionally, Ding et al. [24] propose an overload and unbalance load detection system composed of load cells that automatically detects loading weight, total weight, front and rear unbalanced load, and left and right unbalanced load. Qing et al. [25] designs an innovative detection system of cargo loading position on vehicles based on CCD (Charge-Coupled Device) image sampling equipment. These examples show different types of approaches for unbalanced load detection, but to the knowledge of the authors, none of them use unsupervised automatic methodologies.

Recent technological advances allow the development of automatic methodologies for detecting and classifying different types of damage in railway systems. These methodologies rely on advanced signal processing combined with machine learning techniques and are typically based on the extraction of proper features to distinguish undamaged and damaged situations. Previous studies demonstrate good results in railway defect detection using these approaches, namely, in the detection of train wheel damages, such as flats [26,27,28,29,30,31], out-of-roundness [32], and squats and corrugation [33]. Typically, these damage identification techniques require several operations including [34,35]: (i) data acquisition, (ii) feature extraction, (iii) feature normalization, (iv) feature fusion, and (v) feature classification. Other innovative techniques for wheel flat fault detection include a time–frequency ridge estimation method [36] or a multiscale morphology analysis [29] that can effectively extract the influential features of signals from strong background noise and under variable speed conditions.

After acquiring data from the wayside monitoring systems, feature extraction is performed. This operation converts the time-series data into more condensed information, allowing damage to be more easily observed. Symbolic data, continuous wavelet transform [27,37], principal component analysis (PCA) [38,39], and autoregressive models [34,40] are examples of effective techniques for extracting damage-sensitive features for both static and dynamic monitoring. In applications where the measured quantity is acceleration, autoregressive models (AR) have been widely disseminated, and the autoregressive model with exogenous input (ARX) [35] shows a higher sensitivity to damage in relation to AR due to its ability to capture cross information between sensors.

The feature normalization is an important operation because it allows removing the environmental and operational effects (EOV’s) from the identified dynamic response. The objective of this procedure is to obtain normalized features sensitive to the damage, and typically, latent variable methods are used, such as principal components analysis (PCA) [41] or regressive methods, as well as multiple linear regression (MLR) [42].

In continuation, features fusion operation allows data compression without losing relevant information. The most used algorithms are kernel-based methods [43], manifold learning methods [44], neural networks [45], neighbourhood preserving embedding [46], and Mahalanobis distance [34].

The final operation of the damage detection methodology is feature classification. Typically, two different approaches are used: (i) unsupervised methods, in which models are trained using labelled data under the supervision of training data, such as k-means [34], self-organizing maps (SOM) [47], and cluster analysis [35], or (ii) supervised methods, in which models are not supervised using training dataset, such as Naive Bayes classifiers [48] and k-Nearest Neighbour classifier [49].

The present research aims to design a low-cost monitoring system capable of detecting situations of instability in railway circulation, such as unbalanced loads. The proposed methodology is based on the dynamic rail responses obtained from accelerometers and strain gauges installed on the track. In this research, two different feature extraction methods are tested. The first one consists of a highly sensitive autoregressive model, the ARX model, and the second is based on the PCA method, to transform the time-series measurements into fault-sensitive features. A feature normalization step is afterward implemented by applying the latent variable method PCA. To improve unbalanced loads detection and classification, a three-stage fusion process is defined. In the first and second levels, all features from the same-type sensors are merged, and in the third level, the different measured quantities (accelerations and strains) are fused to enhance the sensibility to the unbalances. The last step is feature discrimination which is achieved by performing an outlier analysis to detect unbalances and a cluster analysis to classify them. 

Thus, this numerical study intends to give innovative contributions to the development of an unsupervised AI-based methodology to automatically detect and classify longitudinally and transversely unbalanced loads from evenly loads, with special emphasis on the following aspects:-This automatic AI-based methodology is an improvement regarding the common methods found in literature, based on direct observations and contact force calculations;-This methodology is based on responses obtained not only from strain gauges but also from accelerometers. The use of accelerometers in WIM systems is not so common but brings some advantages. One is the ease of installation, and another is the possibility of being used in a wider monitoring system to detect other types of vehicle damages;-The proposed methodology is tested regarding the number and type of sensors showing a very high accuracy even with a reduced number of sensors.

## 2. Modelling

### 2.1. Freight Vehicle

The vehicle used in this study is a five-freight wagon of the Laagrss type, whose numerical model was developed by Bragança et al. [50]. The Laagrss wagon is used for container transportation, as depicted in Figure 1a. Each wagon has a tare weight of 13.5 t, and a maximum overload of 30 t with one container layout, or 2 × 15 t in a two-container layout. The three-dimensional (3D) multibody dynamic model, Figure 1b, was developed in ANSYS^®^ [51] using spring-dashpot elements to simulate the suspensions in all the directions and mass point elements to represent the mass and inertial effects in the centre of gravity of each component of the cars (carbody, bogies, and wheelsets). Rigid beam elements are adopted to link the aforementioned elements.

The mechanical properties of the vehicle are presented in Table 1, namely mass (m) and rotary inertias (I), stiffness (k) and damping (c) of the suspensions, as well as the longitudinal (L), transverse (W) and vertical (H) geometric distances. The subscripts cb and w indicate that the respective property is related to the carbody and wheelsets, respectively. The freight wagon has only one level of suspension, without any bogie, and thus, the wheelset is connected directly to the carbody through the suspension.

### 2.2. Track

The track is modelled in ANSYS^®^, Canonsburg (PA), USA [51] with a multiple-layer scheme, in which the ballast, sleepers, and rails are linked through elastic elements that represent the mechanical behaviour of the interface, as shown in Figure 2. Figure 2a schematically presents the track, while Figure 2b represents the track model in finite elements. The ballast mass is simulated by discrete mass point elements, while the sleepers and rails are modelled with beam elements with appropriate properties. The interfaces, namely, the ballast and pads/fasteners, located below and above the sleepers, respectively, have been considered by using spring-dashpot elements in the three directions. Finally, spring-dashpot elements are used to consider the foundation’s flexibility. The properties of the track model, including the description of the variables presented in Figure 2, are presented in Table 2.

Four artificial irregularity track profiles are taken into consideration in this study. Based on accurate data, power spectral density (PSD) curves are generated and then artificial irregularity profiles in vertical and transverse directions are created in the MATLAB^®^ software 2018a version [52]. Wavelengths between 1 m and 75 m with a sampling discretization of 0.01 m are considered, covering the wavelength ranges D1 (3 m to 25 m) and D2 (25 m to 70 m) defined by the European Standard EN 1384-2 [53]. The four different vertical irregularity profiles (1 to 4) created for both right and left rails are presented in Figure 3. The generated irregularity profiles show amplitude peak values between 5–6 mm, which are lower than those indicated in the European Standard [53].

### 2.3. Vehicle–Track Interaction

The numerical vehicle–track dynamic interaction simulations are carried out with the in-house software VSI—Vehicle-Structure Interaction Analysis validated and described in detail by Montenegro et al. [54] and used in distinct applications [34,55,56]. The key point of this model lies in the wheel–rail contact element connecting the two subsystems. This element governs the contact interface, more precisely the evaluation of the wheel–rail contact forces in the normal and tangential directions. After determining in each time step the position of the contact point, the algorithm computes the normal contact force through the Hertz nonlinear theory [57] and the lateral and longitudinal tangential creep forces through the USETAB routine [58]. This numerical tool is implemented in MATLAB^®^ [52] and imports the structural matrices from both the vehicle and track previously modelled in a finite element (FE) package, which in this study is ANSYS^®^ [51]. Note that, although the models of both subsystems are firstly modelled separately, the VSI software links them through a fully coupled methodology. The illustrative approach of the VSI tool is shown in Figure 4.

## 3. Simulation

The automatic unbalanced loads detection methodology is tested and validated in this research with numerical dynamic rail response that simulates specific and known scenarios in terms of load scheme. For testing the proposed detection methodology, scenarios in normal operation conditions (considered into a baseline) and unbalanced scenarios are considered. The unbalanced limits defined in the UIC code [6] are used to generate unbalanced load schemes for the freight wagon and also as a limit to separate permissible and impermissible unbalance loadings. In the case of two-axle wagons, for longitudinal unbalances, the ratio of load per axle should be less than 2:1, and for transverse unbalances the ratio of load between the wheels (left/right) of a given axle should be less than 1.25:1.

### 3.1. Virtual Wayside Monitoring Device

The virtual wayside monitoring system is considered to detect trains with unbalanced loads above the limits. The system is composed of a set of 8 accelerometers and 8 strain gauges mounted on the rail at mid-span between two sleepers, as illustrated in Figure 5. The numbers 1 to 8 in the figure represent the positions of the measurement points, in the right and left rails. In order to have a more realistic track response, an artificial noise of 5% is considered in the numerical measurements. To obtain a closer and more reliable reproduction of the dynamic response and to the sampling frequency of real sensors, a sampling frequency equal to 10 kHz is used.

### 3.2. Baseline

In order to define a robust baseline with a broad range of scenarios for detecting situations of instabilities, several simulations are conducted with different configurations of loads, track irregularities, and vehicle speeds. Figure 6 summarizes the assumptions for baseline scenarios, using six load schemes, four irregularity track profiles (1–4), and five different speeds (40 to 120 km/h in intervals of 20 km/h). A total of 113 simulations are performed with the Laagrss-type freight train composed of five wagons. The considered loading scenarios are empty, half-load, and full load plus three other loading schemes (UNB1, UNB2, and UNB3) with unbalanced loads under the limits prescribed by UIC code of practice [51]. The load schemes UNB1 and UNB2 have unbalanced loads in the longitudinal direction and UNB3 in the transverse direction. The full load scheme presented in Figure 6 corresponds to an even load distribution on the wagon with two containers with 15 t each, without exceeding the maximum mass per axle.

### 3.3. Unbalanced Scenarios

Four different unbalanced loads are defined for a Laagrss-type wagon model above the limits specified by the UIC code of practice [6]. A total of 36 train passages composed of five wagons are considered with the four unbalanced load schemes, which are summarized in Figure 7. The first two correspond to a longitudinal offset of the cargo gravity centre with an unbalance ratio of 2.5 (Long-1) and 3.1 (Long-2), and the other two correspond to a transverse offset of the cargo gravity centre with an unbalance ratio of 1.45 (Transv-1) and 1.70 (Transv-2). In total, 9 analyses are performed for each unbalanced load scheme using the irregularity profile number 2. For each unbalanced load scheme, the unbalanced loadings are simulated in different localizations, namely, in the first, third, and last wagons, and the rest are set with the full load scheme, and for three different speeds, 60, 80, and 100 km/h.

### 3.4. Track Dynamic Response

In Figure 8, time-series of the baseline scenarios in accelerators and strain gauges sensors installed in the rail in position 1 are presented. All time-series are filtered based on a low-pass Chebyshev type II digital filter with a cut-off frequency equal to 500 Hz. These figures show the influence of different loading schemes, train speeds, and irregularity profiles on the track response. Figure 8a,b show that the rail acceleration and strain for full load and UNB2 scheme affect the track responses on the peak acceleration and strain values. Figure 8c exposes the relevant influence of the train speed especially in acceleration records, evidencing the need to consider various train speeds in the baseline. In the case of strain responses, as shown in Figure 8d, the effect of train speed has no significance, as strains are not sensitive to the dynamic component of the train actions. Finally, in Figure 8e,f, two irregularity rail profiles are considered, and the results show that both profiles induce similar rail acceleration and strain responses.

Moreover, some results are shown for unbalanced scenarios at measured position 1. Figure 9a,b compare even full load distribution with longitudinal unbalanced scenario 2, while Figure 9c,d show left and right rail responses considering a transverse unbalanced load scheme. The results show a difference in terms of peak acceleration and strain values during the first wagon passage, where the unbalanced load exists. In the case of the transverse unbalanced load, the acceleration and strain peak values are higher for the right rail than the left rail, a consequence of the load asymmetry. In general, the strain responses are clearer than the acceleration responses, and visually, the unbalance is more distinguishable.

## 4. Methodology for Unbalanced Loads Detection

### 4.1. Overview

The Artificial Intelligence (AI)-based methodology for automatic detection and classification of situations of unbalanced loads is presented in Figure 10 [35,40]. The methodology relies on the accelerations and strains measurements acquired by numerical analysis of baseline and unbalanced train passages in the virtual monitoring system and includes a four-step procedure developed in MATLAB^®^ [52] to be applied to the measured data, particularly:(i)The feature extraction from the measured data is made by testing two different indicator extraction techniques, namely, ARX models and PCA.(ii)To remove the effects of operational and environmental variations, the method of latent variables PCA is adopted.(iii)The Mahalanobis distance is then used for the fusion of all features into one damage index for each simulation but also to combine features from various sources. Thus, in the first level is performed the fusion of all features, in the second level, the fusion of all sensors, and in the third level, the fusion of the different measured quantities, in this case, accelerations and strains.(iv)For feature discrimination, automatic detection is achieved by combining ARX-based features and outliers analysis, and for classification, an automatic clustering process based on the k-means technique and PCA-based features is implemented.

### 4.2. Feature Extraction—ARX vs. PCA

ARX models for feature extraction are applied since they can perform a significant fusion while accurately generalizing the information contained in the data. This is a time-series analysis method that considers the predictive behaviour of the system in a given location considering its history at that same measurement point and the predictive response at other measurement positions in the face of the same event—exogenous inputs. The ARX model can be defined by the equation below:(1)$$x_{j}=\overset{n_{a}}{\underset{i=1}{\sum }}a_{i}x_{j-i}+\overset{n_{b}}{\underset{k=1}{\sum }}b_{k}y_{j-k}+\varepsilon_{j}$$
where *x_j_*, *y_j_*, and *ε_j_* are output, input, and error terms of the model at the signal value *j*, respectively. On the other hand, *n_a_*, *n_b_* and *a_i_*, *b_k_* represent the orders and the parameters of the output and input data, respectively. Both order numbers are considered equal to 40, based on the Aaike Information Criteria [59]. Thus, for each event, the total number of extracted features is 80. The sensor installed in position 5 is used as input and each of the remaining sensors is defined as output.

The PCA method is also used as a feature extractor, and a comparison with the ARX model is performed to test which technique leads to a more efficient methodology. The main idea is to get a dimensionality reduction that identifies important relationships in the data, through a Covariance Matrix, and through a linear transformation into principal components using Eigenvectors. Considering an input data matrix X (*n* × *m*), principal components or scores can be calculated based on the following equation:(2)PC=X·T
where *PC* is the (*n* × *m*) matrix of the principal components or scores, and *T* is the (*m* × *m*) linear orthonormal transformation matrix with each column being an Eigenvector of *X*.

Prior to the application of the PCA method, a standardization of the input data matrix *X* using Z-score, by subtracting the mean and dividing by the standard deviation for each value of each column, should be carried out. It is important to perform standardization before PCA because PCA is quite sensitive regarding the variances of the initial variables. After the calculation of the principal component by using Equation (2), four statistical parameters are extracted from the PCA scores, namely, the root mean square (RMS), the standard deviation (SD), the skewness, and the kurtosis. Thus, with this method, the number of extracted features is reduced to 4.

For each sensor, the extracted features constitute a matrix of *n* × *m*, where *n* is the number of different train passages related to the baseline or unbalanced scenarios, and *m* represents the number of extracted features. The dimension of matrices *X* in each method is:-Matrix *X*_ARX_ with 149 × 80, ARX features;-Matrix *X*_PCA_ with 149 × 4, PCA features.

Figure 11 depicts four features relating to each technique mentioned above, considering the accelerometer 1. The extracted features attained with ARX are represented in Figure 11a,b, while the extracted features by using PCA are represented in Figure 11c,d. The results are divided into two groups: baseline scenarios (113) and unbalanced scenarios (36) subdivided into longitudinal and transverse unbalanced loads with two magnitudes as defined in Figure 7.

In Figure 11a,d, the features present a greater dispersion in baseline and unbalanced scenarios, which is indicative of the influence of the effects of operational variations considered in the simulated scenarios, and the difference of amplitude for baseline and unbalanced passages is almost imperceptible. In turn, Figure 11b,c show, for feature 45 of ARX and feature 2 of PCA, that the amplitude variation of the unbalanced scenarios is greater than the baseline scenarios. This proves that only specific features have the potential to identify unbalanced loads. Particularly for PCA, feature 2, related to the standard deviation, shows sensitivity to distinguish between longitudinal and transverse unbalanced loads.

### 4.3. Feature Normalization—PCA

As previously mentioned, the second step of the AI methodology is feature normalization, and the PCA method is used to eliminate the effects of the environmental and operational variations (EOV’s) from the measured responses. With this procedure, the features show greater sensitivity to damage. Assuming these EOV’s have a linear effect on the identified parameters, the PCA method can efficiently remove the results of EOV’s. Moreover, the most relevant information related to the operational and environmental effects is retained in the first axes of the PCA. The number of p components to be discarded is calculated by the common rule that considers the percentage of total variance explained by each component to be equal to 80% [27]. In both PCA and ARX features, only the first principal component is discarded. Figure 12 represents the two normalized features after applying the PCA normalization to the ARX and PCA features.

In the case of ARX features, as shown in Figure 12a,b, after removing EOVs, the features remain sensitive to the unbalanced loads, and for feature 2, a significant variation of amplitude occurs between baseline and scenarios with load instability. For PCA features, as shown in Figure 12c,d, after normalization, only feature 2 shows a variation of amplitude between baseline and unbalanced scenarios. Moreover, this feature is sensitive to distinguish between longitudinal and transverse unbalanced loads.

### 4.4. Data Fusion—Mahalanobis Distance

After feature normalization, a data fusion is performed to increase the sensitivity of the features to the abnormal cases, and as a result, a damage index (DI) is achieved for each simulation. Data fusion aims to reduce the volume of the extracted data by persevering the most relevant information, that is, preserving or even enhancing the ability to characterize unbalanced scenarios.

Mahalanobis distance (MD) is used to reduce multivariate data into one single DI [60]. The MD calculates the distance between the unbalanced and baseline scenarios to define similarities between them, where short distances represent large similarities. The MD is generic enough to be used to detect any unbalanced scenario, while providing a weighting that is entirely unsupervised, and therefore independent of human intervention, type of train, etc. It consists of a weighted DI in which the weights are determined by the covariance structure. In addition, and more importantly, the weighting proportional to the covariance structure provides an additional layer of feature normalization which allows outlining with high sensitivity those that were not used for the definition of the covariance structure.

The MD, named herein as DI, is calculated for each simulation by the following expression:(3)$$DI=\sqrt{\left(x_{i}-\bar{x}\right)\cdot S_{x}{}^{-1}\cdot {\left(x_{i}-\bar{x}\right)}^{T}}$$
where:

$x_{i}-$Matrix with potential damage features;

$\bar{x}—$Matrix with the mean of estimated features in baseline scenario;

$S_{x}—$Matrix with the covariance of baseline scenario simulations.

To merge all the features attained with the ARX and PCA techniques, the MD is implemented for each of the sensors. In a first level, the fusion of all features is performed, and in a second and a third levels, the fusion of all sensors is performed for the different measured quantities, respectively. The first fusion process results in a 149 × 1 vector for each one of the 8 sensors for each technique. Figure 13 shows the results of this 1st level of data fusion based on each of the techniques (ARX and PCA) relative to accelerometer 1. It is possible to observe that for ARX features, this process of fusing improves the detection sensitivity as a clear distinction between the baseline and unbalanced loads as depicted in Figure 13a.

To further increase the sensitivity or distinction between baseline and unbalanced scenarios, a second level of fusion—sensors fusion—is implemented. This process culminates in a vector with 173 × 1 in which all the features from each type of sensor are merged. In a subsequent step, the obtained vector from the second level is standardized using Z-score. Then, in a third level of fusion, the features from the two types of sensors, accelerometers, and strain gauges are merged. At the end of this fusion process, the output is one feature vector that combines the information from all types of sensors.

This single feature vector increases the amplitude difference between the baseline and unbalanced scenarios, increasing the possibility of effective detection of unbalanced loads, as shown in Figure 14. By comparing Figure 13 and Figure 14, the minimum amplitude values increase from around 12 to 30, for ARX features, and from near zero to 10, for PCA features. Moreover, the application of this third level of fusion to PCA features has enhanced the difference between the longitudinal unbalances (Long-1 and Long-2) and the transverse unbalances (Transv-1 and Transv-2).

After the fusion stage, it can be concluded that the most promising method for unbalanced load detection is based on the ARX features because the amplitude difference between baseline and unbalanced scenarios is significantly higher than the one obtained by PCA features. For classification, given that only PCA features show sensitivity to distinguish longitudinal and transverse unbalanced scenarios, the considered method should be based on PCA features. Based on these conclusions, in the next step of feature discrimination, the detection method is based on ARX features, and the classification method is based on PCA features.

### 4.5. Feature Discrimination

#### 4.5.1. Outlier Analysis

Feature discrimination is the last phase of the AI-based methodology for the automatic detection of unbalanced loads. In the proposed methodology, the outlier analysis is adopted for unbalanced loads detection using ARX features. Based on the obtained results with ARX features using the second and third level of data fusion (Figure 14a), a confidence boundary (CB) is applied to distinguish between baseline and unbalanced scenarios. The CB is calculated by using the Gaussian inverse cumulative distribution function (ICDF), considering the mean value, *ū*, and standard deviation, *σ*, of the baseline feature vector, and for a significance level α:(4)$$CB=invF_{x}\left(1-\alpha \right)$$
where
(5)F(x| μ¯,σ)=1σ2π∫−αxe−12(x−μ¯¯σ)2dy,with xϵℝ

Consequently, when DI is equal to or higher than CB, a feature is considered to be an outlier. The threshold significance level is defined as 1%, as it is normally observed in several structural integrity monitoring studies to identify damage [34,61].

Figure 15 shows the effectiveness of the proposed methodology by comparing the CB (represented as a green line) with the different DIs of each one of the 36 train passages with unbalanced loads. Since the DIs for the first 113 passages are less than CB, the algorithm can successfully distinguish an evenly train loading from an unbalanced one. As proved by the different plots of Figure 15, unbalanced load detection is successfully achieved, with no false-positive or false-negative cases, by using 8 accelerometers (Figure 15a), 8 strain gauges (Figure 15b), or by using 8 accelerometers together with 8 strain gauges (Figure 15c).

#### 4.5.2. Cluster Analysis

For feature classification, a clustering process is proposed that aims to divide data sets into different clusters, which must be as compact and separate as possible. In this study, the k-means clustering technique is adopted, by using the city-block distance. The purpose of clustering is to regroup the data into separated *k* clusters based on their feature vector distances. The k-means algorithm is defined as follows: given a data vector (features) and some arbitrary *k* clusters, the idea is to find *k* centroids that minimize the distance of each vector to its nearest centroid. As previously mentioned, the k-means clustering method requires the number of clusters to be defined in advance. To tackle this limitation, the global silhouette index (SIL) [35] is used, making this an automatic process. Based on the obtained results with PCA features using the second and third-level data fusion, Figure 16 shows the obtained clusters for each type of sensor. 

The plots demonstrate that the k-means method can distinguish different combinations of unbalanced cases when using the PCA feature set. By using the accelerometer responses (a total of 8 sensors), in the case of Figure 16a, it is possible to separate the unbalanced scenarios into two large clusters, classifying the PCA features into longitudinal and transverse unbalanced cases. When using the strain gauges responses (a total of 8 sensors), as shown in Figure 16b, the robustness of the classification increases, without any false classification. By using the responses from 16 sensors (8 accelerometers plus 8 strain gauges), as depicted in Figure 16c, it is possible to efficiently separate the unbalanced scenarios into three clusters, one comprising all longitudinal cases (long 1 and Long-2), another one with the less severe transverse case (transv-1), and a last one with the more severe transverse case (transv-2).

#### 4.5.3. Sensitivity for Different Sensor Layouts

The presented automatic procedure is applied for the accelerometers and strain gauges responses of the 16 sensors positioned in the mid-span rail. To evaluate the optimized number of sensors used to detect and classify unbalanced loads, a sensitivity analysis is performed.

The results for three different numbers of sensors are plotted in Figure 17 for outlier (ARX-based features) and cluster analysis (PCA-based features).

In Figure 17a, the results using a 16-sensor configuration, with 8 accelerometers and 8 strain gauges, which is the configuration used in the results shown in previous sections, are presented. Then, in Figure 17b,c, a reduced number of sensors are considered respectively for an 8-sensor configuration (4 accelerometers and 4 strain gauges) and a 4-sensor configuration (4 accelerometers and 4 strain gauges). In terms of outlier analysis, the DI amplitude decreases with the reduction of the number of sensors. However, in all sensor configurations the methodology shows to be very efficient in detection. Regarding the cluster analysis, the well-defined three clusters obtained with 16 sensors start to diffuse into two distinct clusters in the case of 8 sensors, losing efficiency for the 4 sensors configuration.

Table 3 summarizes the efficiency results of the proposed methodology in terms of detection and classification for different sensor configurations, using accelerometers and strain gauges. The results show a very good efficiency of the methodology in terms of detection, with a negligible number of false positives and with no false negatives. It is possible to conclude that although only two accelerometers or two strain gauges proved to be enough to detect unbalanced loads, it is not sufficient to classify the unbalanced scenarios as longitudinal or transverse.

From an economic point of view, 8 sensors, consisting of 4 accelerometers and 4 strain gauges, are able to detect and classify longitudinal and transverse unbalanced loads. However, it should be highlighted that only two sensors are sufficient to detect unbalanced loads automatically. As transverse unbalanced loads tend to be more severe than longitudinal unbalances, classification is very helpful for railway operators, to control safety against derailment of the vehicles. The PCA-based classification process shows that it can efficiently separate the longitudinal from transverse unbalanced cases, being the transverse case the one with the higher amplitude.

## 5. Conclusions

This paper proposes an automatic methodology based on artificial intelligence that can accurately identify unbalanced vertical axle load situations which can put at risk the running safety of the operating freight trains.

Two different feature extraction methods are tested, one consisting of a highly sensitive autoregressive model, the ARX model, and the other one based on the PCA method to transform the time-series accelerations and strains into sensitive features for unbalanced load detection.

The ARX method proved to be the most effective feature extraction method for detecting unbalanced loads, as the distance between the unbalanced scenarios and the confidence boundary is high enough to have no false unbalanced identification. The extracted features based on the PCA method showed to be more sensitive to the transverse unbalanced load cases and proved to be very robust for the classification between longitudinal and transverse unbalanced scenarios.

The effectiveness of the proposed methodology is tested regarding the number and type of sensors with very good accuracy. Both acceleration and strain data responses provided good performance in detection. Although the installation of only two accelerometers or two strain gauges proved to be adequate to detect an unbalanced load, it is not enough to classify the unbalance according to the type (transverse or longitudinal). From an economic point of view, the optimal layout solution for detecting and classifying longitudinal and transverse unbalanced loads consists of a wayside system with 4 accelerometers and 4 strain gauges installed symmetrically on both rails.

Future works include experimental tests to validate the proposed methodology based on on-site measurements. Furthermore, the upgrade of the current methodology to localize unbalanced loads in multiple wagons of a freight train will be considered.

## Figures and Tables

**Figure 1 sensors-23-01544-f001:**
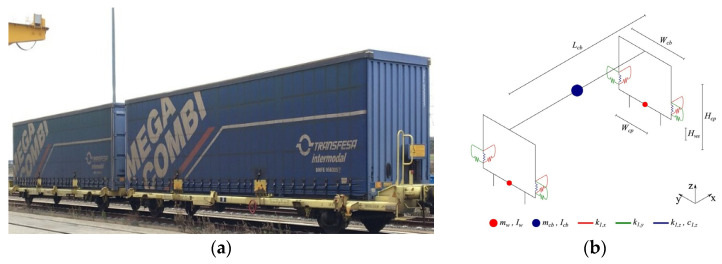
Laagrss type freight wagon: (**a**) double wagon overview; (**b**) numerical model.

**Figure 2 sensors-23-01544-f002:**
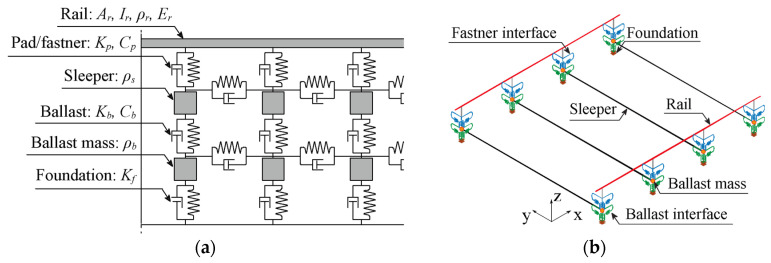
Numerical model of the track: (**a**) multi-layer representation and (**b**) FE model.

**Figure 3 sensors-23-01544-f003:**
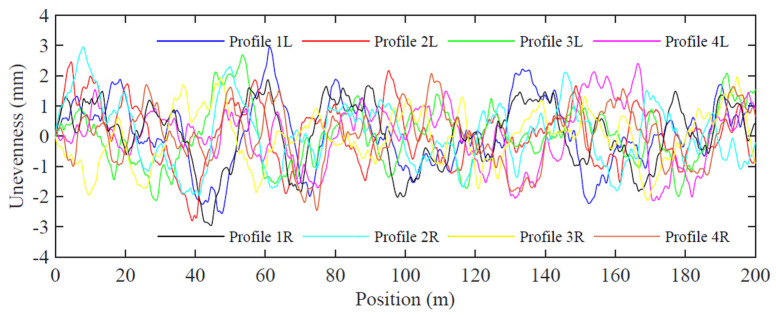
Longitudinal irregularity profiles (1–4) for left and right rails.

**Figure 4 sensors-23-01544-f004:**
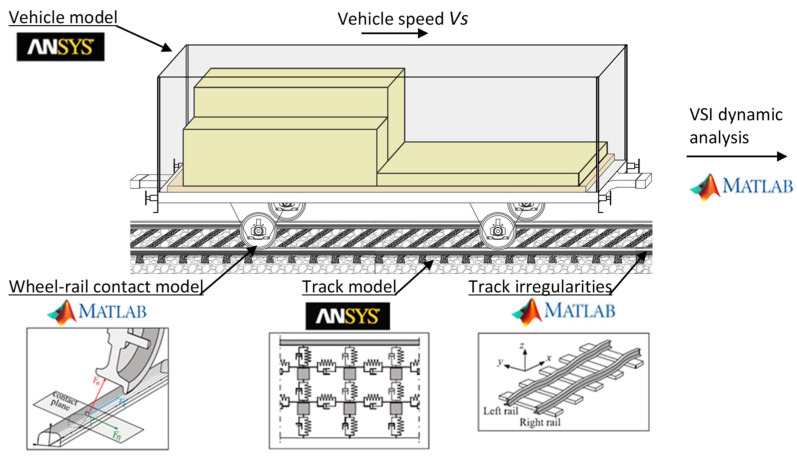
Vehicle–track interaction model schematization.

**Figure 5 sensors-23-01544-f005:**
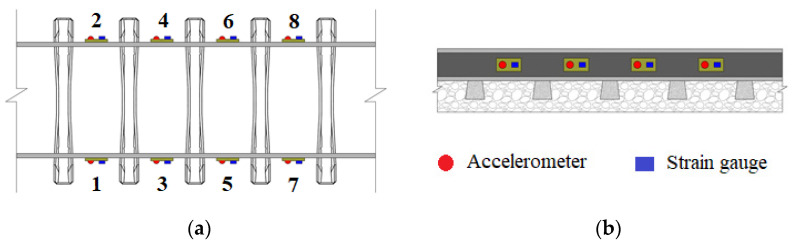
Virtual wayside monitoring system: (**a**) top view, (**b**) lateral view.

**Figure 6 sensors-23-01544-f006:**
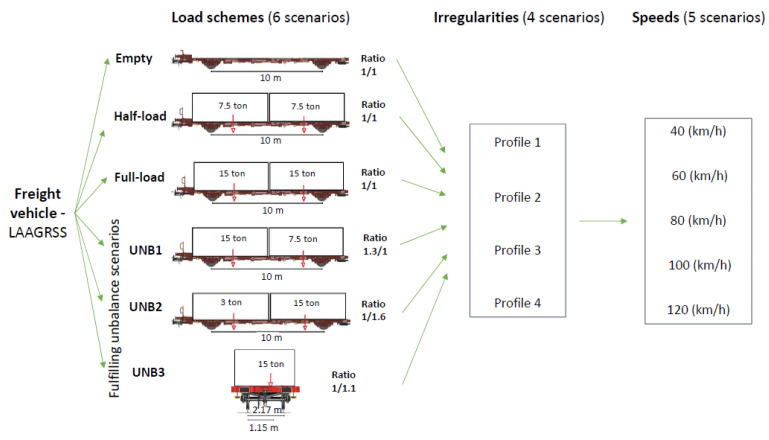
Baseline scenarios.

**Figure 7 sensors-23-01544-f007:**
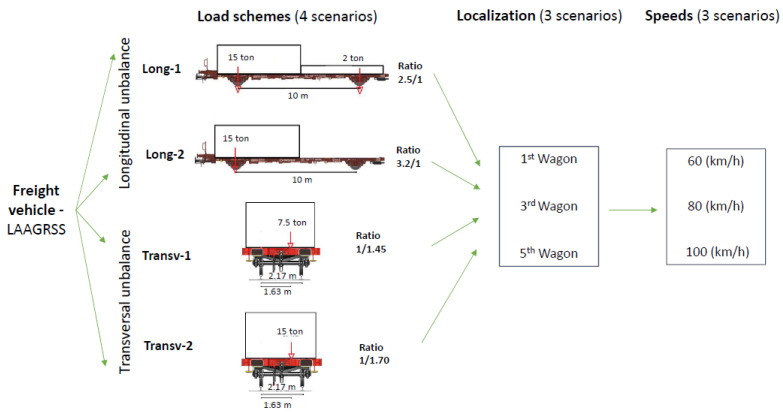
Unbalanced load scenarios.

**Figure 8 sensors-23-01544-f008:**
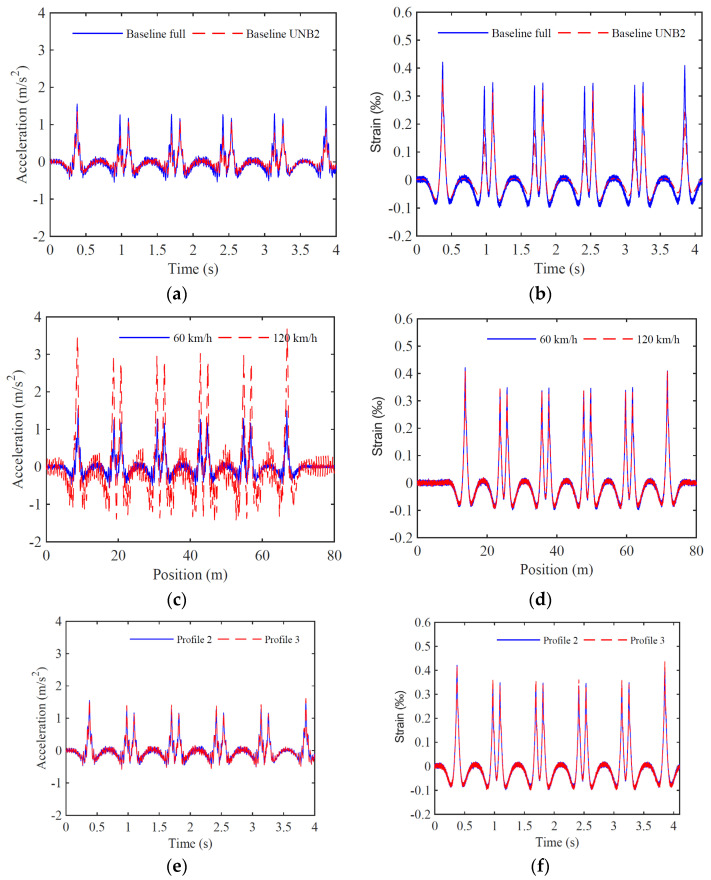
Time-series with accelerations (**a**,**c**,**e**) and strains (**b**,**d**,**f**) measured in position 1 for baseline scenarios: (**a**,**b**) comparison between full and UNB2 load schemes; (**c**,**d**) comparison between 60 and 120 km/h; (**e**,**f**) comparison between irregularities profile 2 and 3.

**Figure 9 sensors-23-01544-f009:**
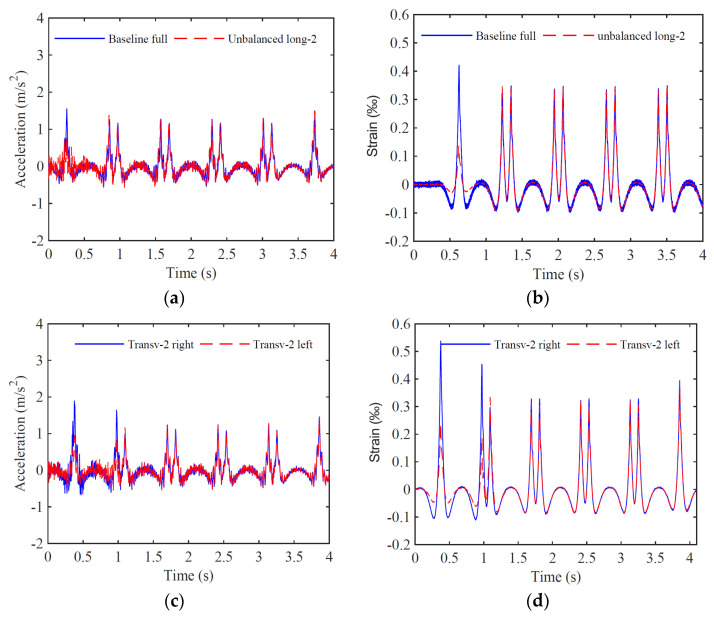
Time-series with accelerations (**a**,**c**) and strains (**b**,**d**) measured in position 1 for unbalanced scenarios: (**a**,**b**) comparison between full and longitudinal unbalanced load scheme; (**c**,**d**) comparison between a transverse unbalance recorded in right and left rails.

**Figure 10 sensors-23-01544-f010:**
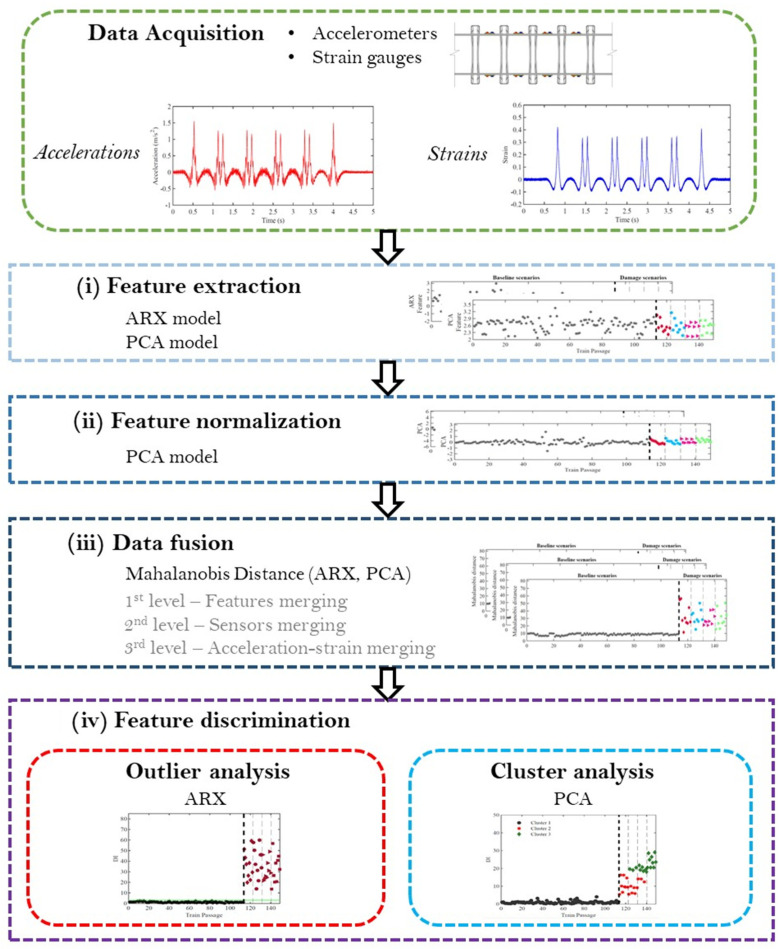
Methodology for the automatic identification of unbalanced loads.

**Figure 11 sensors-23-01544-f011:**
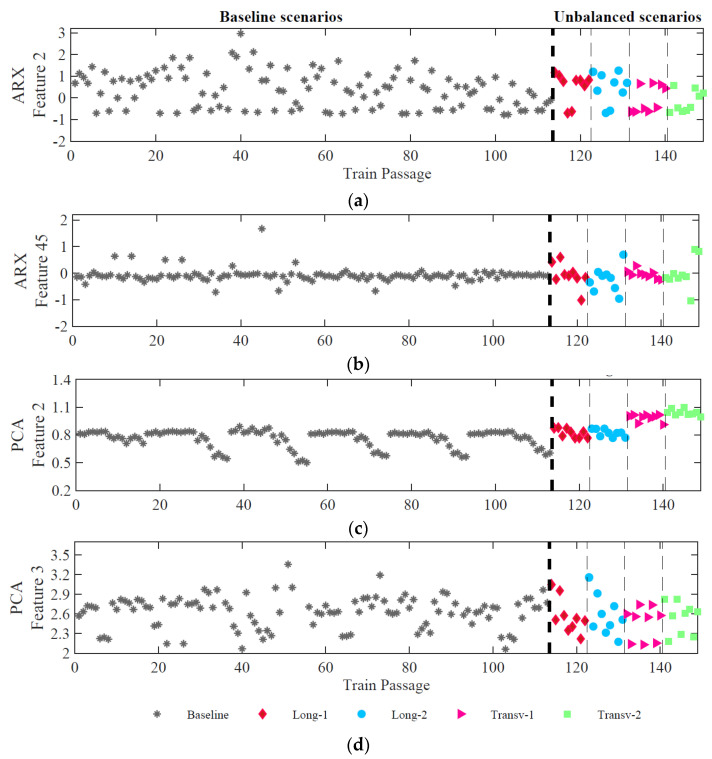
Feature extraction for accelerometer 1 baseline and unbalanced scenarios: (**a**) ARX model, feature 2; (**b**) ARX model, feature 45; (**c**) PCA model, feature 2; (**d**) PCA model, feature 3.

**Figure 12 sensors-23-01544-f012:**
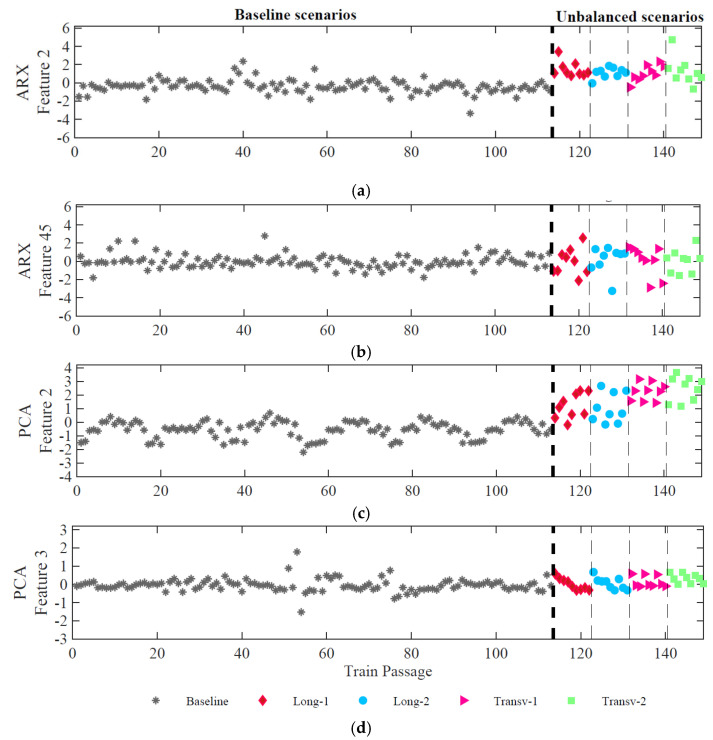
Feature normalization for accelerometer 1: (**a**) and (**b**) based on ARX features; (**c**) and (**d**) based on PCA features.

**Figure 13 sensors-23-01544-f013:**
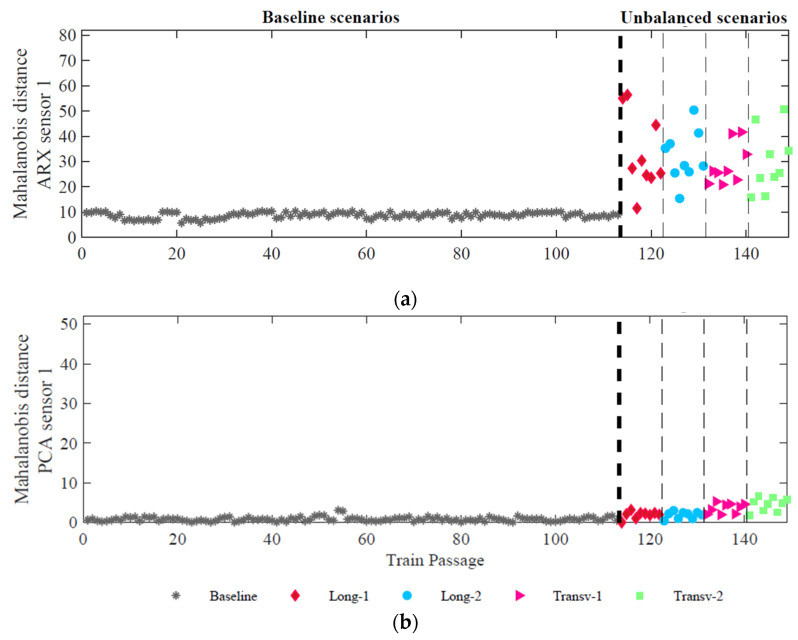
1st level data fusion—for accelerometer 1: (**a**) ARX features; (**b**) PCA features.

**Figure 14 sensors-23-01544-f014:**
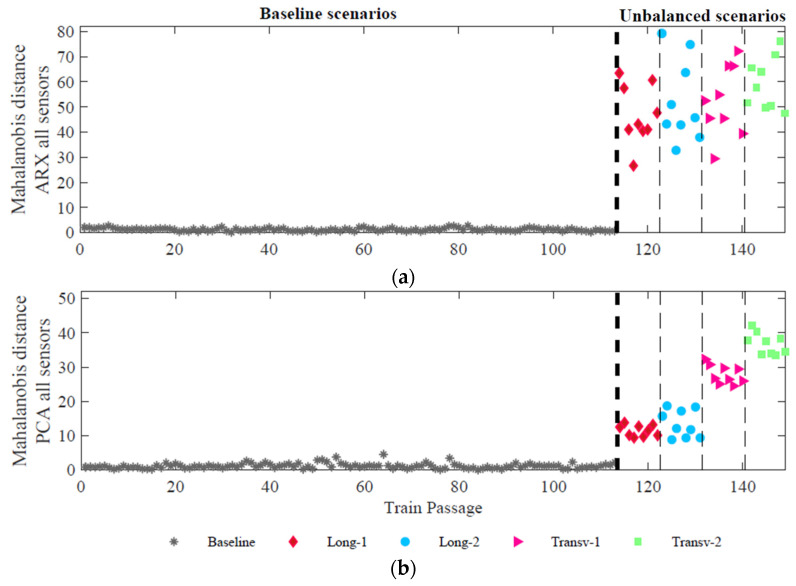
Data fusion 3rd level—merging of all sensors (strains and accelerations) (**a**) ARX features (**b**) PCA features.

**Figure 15 sensors-23-01544-f015:**
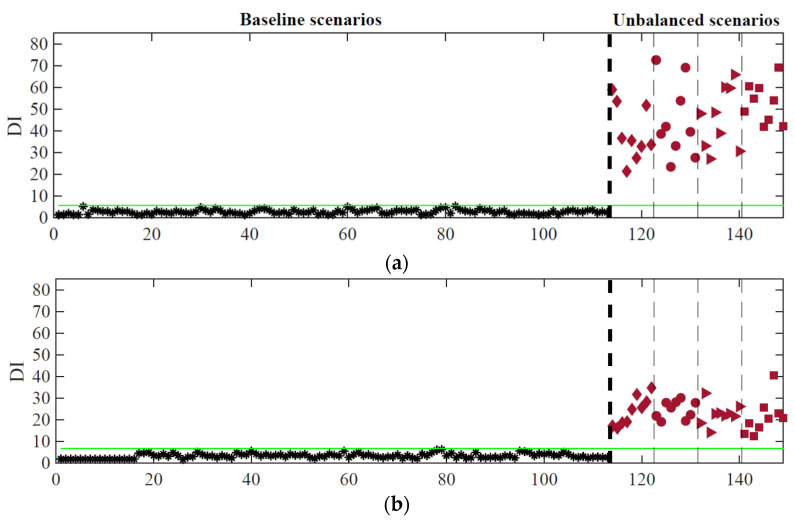
Automatic unbalanced loads detection based on ARX features considering the responses from: (**a**) 8 accelerometers; (**b**) 8 strain gauges; (**c**) 8 accelerometers + 8 strain gauges.

**Figure 16 sensors-23-01544-f016:**
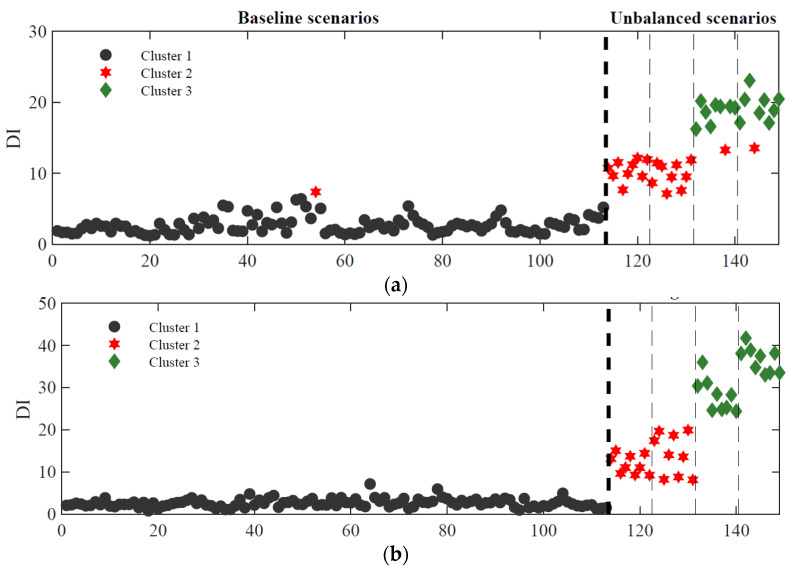
Cluster analysis for the PCA features: (**a**) 8 accelerometers; (**b**) 8 strain gauges; (**c**) 8 accelerometers + 8 strain gauges.

**Figure 17 sensors-23-01544-f017:**
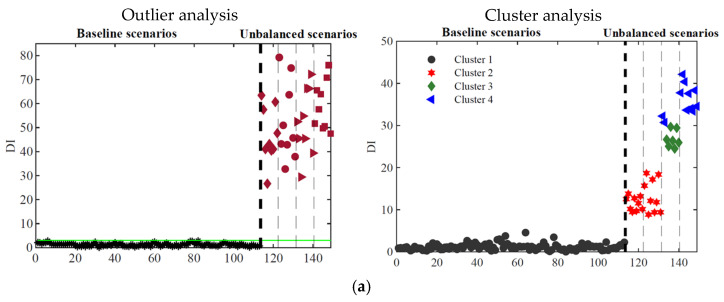
Sensitivity analysis with respect to the number of sensors (accelerometers + strain gauges): (**a**) 16 sensors (8 + 8); (**b**) 8 sensors (4 + 4); (**c**) 4 sensors (2 + 2).

**Table 1 sensors-23-01544-t001:** Mechanical properties of the freight vehicle model.

Carbody		Wheelset		Suspensions	
Parameter	Value	Parameter	Value	Parameter	Value
Mass $m_{cb}\;\left(\mathrm{ton}\right)$	13.5	Mass $m_{w}\;\left(\mathrm{kg}\right)$	1247	Longitudinal stiffness $k_{1,x}\;\left(\mathrm{kN}/m\right)$	44,981
Roll moment of inertia, $I_{cb,x}\;\left(t.m^{2}\right)$	49	Roll moment of inertia $I_{w,x}\;\left(\mathrm{kg}.m^{2}\right)$	312	Lateral stiffness $k_{1,y}\;\left(\mathrm{kN}/m\right)$	30,948
Pitch moment of inertia, $I_{cb,y}\;\left(t.m^{2}\right)$	673	Yaw moment of inertia $I_{w,z}\;\left(\mathrm{kg}.m^{2}\right)$	312	Vertical stiffness $k_{1,z}\;\left(\mathrm{kN}/m\right)$	1860
Yaw moment of inertia $I_{cb,z}\;\left(t.m^{2}\right)$	665			Vertical damping $c_{1,z}\;\left(\mathrm{kN}.s/m\right)$	16.7
Length $L_{cb}\;\left(m\right)$	10				
Height $H_{cb}\;\left(m\right)$	2.17				
Width $W_{cb}\;\left(m\right)$	2.297				

**Table 2 sensors-23-01544-t002:** Stiffness properties of the railway track.

Parameter		Value
Rail	*Ar* (m^2^)	7.67 × 10^−4^
	*ρ_r_* (kg.m^3^)	7850
	*I_r_* (m^4^)	30.38 × 10^−6^
	*E_r_* (N/m^2^)	210 × 10^9^
Rail pad	*K_p_* (N/m) (*Kx/Ky Kz)*	20 × 10^6^/20 × 10^6^/500 × 10^6^
	*C_p_* (N.s/m) (*Cx/Cy Cz)*	50 × 10^3^/50 × 10^3^/200 × 10^3^
Sleeper	*ρ_s_* (N/m)	2590
Ballast	*Kb* (N/m) (*Kx/Ky Kz)*	900 × 10^3^/2250 × 10^3^/30 × 10^6^
	*Cb* (N/m) (*Cx/Cy Cz)*	15 × 10^3^/15 × 10^3^/15 × 10^3^
Foundation	*Kf* (N/m) (*Kx/Ky Kz)*	20 × 10^6^/20 × 10^6^/20 × 10^6^

**Table 3 sensors-23-01544-t003:** False detection and classification relating to the number of sensors.

Sensors (Accelerometers + Strain Gauges)	Detection		Classification		
	False Positives	False Negatives	Cluster	Groups	False Classifications
16 (8 + 8)	0% (0/113)	0% (0/36)	3	(Long 1-2) (Transv1) (Transv2)	6% (2/36)
12 (6 + 6)	0% (0/113)	0% (0/36)	2	(Long 1-2) (Transv 1-2)	6% (2/36)
8 (4 + 4)	0% (0/113)	0% (0/36)	2	(Long 1-2) (Transv 1-2)	8% (3/36)
4 (2 + 2)	2% (2/113)	0% (0/36)	2	(Long 1-2) (Transv 1-2)	25% (9/36)
2 (2 + 0)	1% (1/113)	0% (0/36)	2	(Long 1-2) (Transv 1-2)	28% (10/36)
2 (0 + 2)	1% (1/113)	0% (0/36)	2	(Long 1-2) (Transv 1-2)	39% (14/36)

## Data Availability

Not applicable.

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
