# Peer review of "Early Identification of Unbalanced Freight Traffic Loads Based on Wayside Monitoring and Artificial Intelligence"

_sensors, 2023, doi:10.3390/s23031544_

Round 1

Reviewer 1 Report

The paper presents a monitoring method to detect unbalance of the freight trains using vibration and strain signals. The authors should provide a comprehensive review on related studies, such as the fault detection method on railway wheel flat using an adaptive multiscale morphological filter and time-frequency ridge estimation an effective for fault diagnosis at time-varying speeds. Moreover, the sequences of  the references are also improper. For example, Refs. [30-32] are followed by [3]. I also suggest the authors demonstrating their method's performance by experimental signals.

Reviewer 2 Report

The authors proposed automatic methodology based on artificial intelligence that can accurately identify unbalanced vertical axle load situations and present good simulation work.

However, the paper does not mention any similar work in the literature and the performance comparison or improvement is not presented with respect to the methods proposed in literature.

Reviewer 3 Report

The early detection method  is presented to automatically identify unbalanced vertical loads, and then prevent significant damages such as service interruptions or derailments. The research involves feature extraction, feature normalization, data fusion and feature discrimination. There are some innovations in the method, the simulation cases are more comprehensive. However, there are some questions to be asked.

1. Much simulation data are used to verify the proposed method. If real data of unbalanced freight traffic loads are used, there is greater support for this study.

2. How do the DI value of early warning set? 

Round 2

Reviewer 1 Report

The authors have addressed all my queries. The paper can be published in present form.